# Comparing the Impact of Urban Park Landscape Design Parameters on the Thermal Environment of Surrounding Low-Rise and High-Rise Neighborhoods

**Sihan Xue [1,\*], Liang Yuan [1], Kun Wang [2,\*], Jingxian Wang [2] and Yuanfeng Pei [1]**

[1] School of Architecture, Zhengzhou University, Zhengzhou 450001, China; yuanliang233q@163.com (L.Y.); 15649850623@163.com (Y.P.)

[2] College of Landscape Architecture and Art, Henan Agricultural University, Zhengzhou 450002, China; wjxwjx326@163.com

[\*] Correspondence: xuesh@zzu.edu.cn (S.X.); wkun@henau.edu.cn (K.W.)

**Abstract:** Scientific and reasonable planning of urban forests is crucial to mitigate the UHI effect and create a comfortable local climate. This study focused on maximizing the synergistic effect of landscape design parameters (i.e., Landscape Shape Index (LSI), Percentage of Green Area (PGA), Park Area (PA), and Percentage of Water Area (PWA)) through orthogonal experimental design and numerical simulation to improve the regulation of the thermal environment of urban parks in the surrounding blocks. For the neighborhood of low-rise buildings, the influence of PGA was the most significant in the morning with a contribution rate of 50.43%, while PWA was the most influential during midday and evening, with contribution rates of 73.02% and 52.30%, respectively. In contrast, PA was the most influential in the morning with a contribution rate of 62.11% for the high-rise building neighborhood, while the impact of PWA was the most significant in the midday with a contribution rate of 43.99%. In addition, the contribution of PA and PWA played vital roles at night. This study proposed combinations of landscape design parameters for urban parks that met the requirements of two types of neighborhoods, which would help guide the planning and construction of urban forests.

**Keywords:** urban forest; thermal environment; landscape design parameters; orthogonal experimental design; numerical simulation

## 1. Introduction

Rapid urbanization in developing countries has resulted in urban heat island issues [1,2]. Hence, this phenomenon has become a critical topic in urban ecology, impacting human health and outdoor activities [3]. During the hot seasons, high-intensity urban heat island areas face significantly increased mortality risks, almost double compared to those of low-intensity heat island areas [4]. Therefore, urgent measures are required to tackle urban heat island problems [5]. Effective use of urban forest resources is crucial for reducing heat island effects and creating healthy and comfortable local climates.

Urban forest resources and thermal environments are closely related, and the scientific guidance for constructing or renovating urban forest spaces has become a prominent research topic [6–8]. In 2020, Michael et al. assessed the impact of green space on the thermal environment of Hackett Port in Nigeria, confirming the positive role of urban forest space in mitigating urban heat [9]. Studies consistently show that higher vegetation coverage leads to more significant cooling effects [10–12], and increasing green areas helps alleviate the urban heat island effect [13,14]. Conversely, decreased green coverage reduces the area with the lowest temperatures within urban forest spaces [15]. Physiological characteristics of plants, such as tree species, height, crown width, and leaf area index, also influence the urban forest cooling effect and the surrounding thermal environment [16,17].

Water bodies contribute to the thermal environment of urban forest spaces and their surroundings [18–20], exhibiting a notable cooling effect [21,22]. The cooling effect of urban forest spaces is influenced by factors such as the percentage of water bodies within them [23–25] and the water shape index [26]. The surrounding landscape configuration also affects the cooling effect, as nearby building sites may weaken the cooling effect of water bodies, while vegetation can expand the cooling range [27]. Combining vegetation and water elements in urban forest spaces demonstrates superior thermal mitigation compared to combinations of vegetation with impermeable surfaces or water elements with impermeable surfaces [28]. Apart from natural landscape elements, the Landscape Shape Index (LSI) and urban forest size also influence the cooling effect of urban forest spaces [6,23,29]. Qiu et al. found that an increased ratio of perimeter to area enables the urban forest's thermal attenuation effect to serve a larger urban area [6]. Jauregui compared temperatures in urban forests of different sizes and observed a temperature difference of 2 °C–3 °C between large-scale urban forests and the surrounding urban environment [30]. Furthermore, the cooling effect of urban forests is affected by the aspect ratios and height-to-width ratios of the surrounding street spaces [31]. It is evident that the cooling effect of urban forests exhibits a scale effect, influenced not only by their landscape design parameters but also by the spatial morphology of the surrounding blocks.

In recent years, remote sensing imagery has been used to study the cooling effect of green spaces on land surface temperature. However, these studies often focus on the cooling effect at a specific time, neglecting the diurnal variation of the cooling effect of green spaces. With advances in computer performance, numerical simulation techniques have become crucial for quantitative prediction and evaluation of microclimates [32]. Simulation studies offer advantages over field measurements as they can eliminate various interfering factors in real-life scenarios. Among the commonly used pieces of computational fluid dynamics (CFD) software [32], ENVI-met stands out for its superior capabilities in assessing the influence of green spaces on microclimates [33,34]. The scientific validity of ENVI-met has been extensively demonstrated, and its comprehensive plant and material databases have made it an essential tool for simulating urban forest microclimates [33]. For instance, Yue et al. [35] found that vegetation plays a significant role in cooling the surrounding environment in their ENVI-met study of urban forests. The average land surface temperature increases as the distance from the vegetation area increases. Additionally, water bodies have a significant impact on improving thermal and humidity conditions within a range of 9–18 m of their surroundings. In another study by Binarti et al. [36], ENVI-met was employed to investigate urban forest spaces, revealing a correlation between tree density and thermal comfort. In general, ENVI-met has gained widespread application in current research on the simulation of microclimates in urban forest spaces.

Urban parks, as integral components of urban forests, play a crucial role in promoting sustainable urban development and improving the living environment. Despite several studies, further research is needed to explore the combined effects of park landscape design parameters such as vegetation, water bodies, Landscape Shape Index (LSI), and park area on microclimates and the mechanisms behind temperature reduction. In addition, the influence of parks on surrounding blocks with varying building heights and densities, as well as the temporal patterns of park-induced cooling effects, requires more investigation. Hence, quantitative studies on the microclimate effects of parks in different spatial configurations are crucial to maximizing the potential for climate regulation in urban forest spaces.

This study aims to investigate the climate regulation potential of park landscape design parameters in cold regions of China with intensified urban heat island effects [2,37]. By surveying 124 parks in Zhengzhou and employing orthogonal experimental design and numerical simulation techniques, this study examines the impact of landscape design parameters and their interactions on the thermal environment inside and outside the parks. Specifically, two key aspects will be emphasized: (1) comparing park cooling differences in cold regions of China during summer between low-rise and high-rise blocks over different time periods; (2) investigating the impact and contribution of landscape design elements

and their interactions on temperature differences inside and outside parks and obtaining optimized design strategies for parks under the synergistic effects of landscape elements.

## 2. Materials and Methods

### 2.1. Landscape Design Parameters

2.1.1. Definition of Landscape Design Parameters

Four factors, namely Park Area (PA), Landscape Shape Index (LSI), Percentage of Green Area (PGA), and Percentage of Water Area (PWA), serve as landscape design parameters. PA denotes the size of the park, whereas PGA and PWA represent the proportion of green and water areas, respectively. LSI quantifies the shape complexity by comparing it to a circular shape of equal area. A higher numerical value indicates a more complex shape. The definition of LSI is as follows:

$$LSI = \frac{Pt}{2\sqrt{\pi \times A}} \tag{1}$$

where $Pt$ represents the perimeter of the park (m) and $A$ denotes the park area (m$^2$).

2.1.2. Analysis of Landscape Design Parameters

As shown in Figure 1a, Zhengzhou City is located in central China, positioned between latitude 34°16′ to 34°58′ north and longitude 112°42′ to 114°14′ east, with a cold climate and high summer Ta (Ta > 30 °C) and RH (>50%). The highest average temperature (27.9 °C) occurs in July, and the maximum average humidity (75%) is observed in August [38].

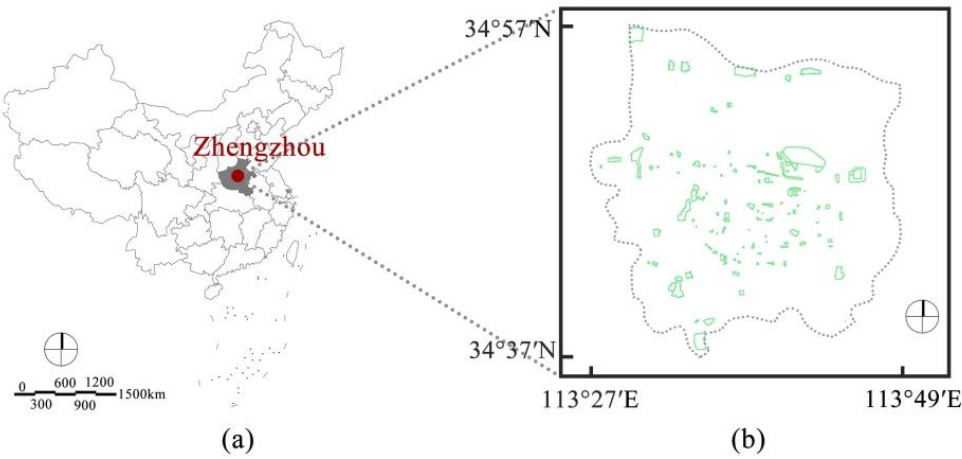

**Figure 1.** Geographic location of the survey area: (**a**) city location; (**b**) survey area. The gray dashed line area represents the urban area. The green areas represent park boundaries within the urban area.

Based on data from 124 surveyed parks (Figure 1b), Table 1 reveals the following findings: Most parks (77.4%) are below 15 hectares in size. Approximately 75% of parks exhibit an LSI ranging from 1 to 1.4. Per the "Standard for Classification of Urban Green space" (CJJ/T 85-2017) [39] and "Code for the Design of Public Park." (GB 51192-2016) [40], green areas (PGA) in different park types should exceed 60%. Among the analyzed cases, 50% of parks have a PGA ranging from 60% to 80%. Parks with a PGA exceeding 80% tend to be larger, mostly surpassing 15 hectares. Water bodies, though not strictly mandated, play a significant role in enhancing the thermal environment. Among surveyed parks, 59.68% lack water bodies, 18.54% have a PWA between 0% and 10%, and 13.70% have a PWA ranging from 10% to 20%. Only 7.25% of parks exceed a PWA of 20%. Furthermore. in park cases with water bodies, the water bodies are mostly located in the central area of the park, surrounded by trees.

**Table 1.** Proportional distribution of park counts across different factor levels.

|  | Number | Proportion |
|---|---|---|
| PA |  |  |
| 0–5 ha | 57 | 45.97% |
| 5–10 ha | 16 | 12.90% |
| 10–15 ha | 23 | 18.55% |
| More than 15 ha | 28 | 22.58% |
| LSI |  |  |
| 0–1 | 6 | 4.84% |
| 1–1.2 | 62 | 50.00% |
| 1.2–1.4 | 31 | 25.00% |
| More than 1.4 | 25 | 20.16% |
| PGA |  |  |
| Less than 60% | 25 | 20.16% |
| 60%–70% | 34 | 27.42% |
| 70%–80% | 28 | 22.58% |
| More than 80% | 37 | 29.84% |
| PWA |  |  |
| 0% | 74 | 59.68% |
| 0%–10% | 23 | 18.54% |
| 10%–20% | 17 | 13.70% |
| More than 20% | 9 | 7.25% |

*2.2. Orthogonal Experiment*

2.2.1. Experimental Design

Based on Section 2.1.2, this experiment assigned three levels to each morphological factor (Table 2): PA (A)—5 ha, 10 ha, 15 ha; PGA (C)—60%, 70%, 80%; PWA (D)—0%, 10%, 20%. The levels of the three factors were set with equal intervals. Furthermore, when LSI is 1.13, it can form a special shape, namely a square. Meanwhile, considering the common range of LSI values, LSI (B) was set at three levels: 1.13, 1.2, and 1.4. Since each factor has three different levels, the combination of four factors forms 81 experimental scenarios in total. To enhance experimental efficiency, the orthogonal experimental design was adopted. Orthogonal experimental design is a primary method of fractional factorial designs and is commonly used for designing experiments with multiple factors and levels [41]. It can select a representative subset of experiments from the full factorial experiments based on orthogonality, which can significantly reduce the number of experimental scenarios [42].

**Table 2.** Level values of each factor.

| Factor Levels | PA (A) | LSI (B) | PGA (C) | PWA (D) |
|---|---|---|---|---|
| 1 | 5 ha | 1.13 | 60% | 0% |
| 2 | 10 ha | 1.2 | 70% | 10% |
| 3 | 15 ha | 1.4 | 80% | 20% |

Orthogonal table L27 ($3^{13}$) was selected (Table 3) because it allows for experimentation with four factors (PA, LSI, PGA, and PWA), each having three levels, and also considers the effects of three sets of interactions. It resulted in 27 experimental scenarios, reducing the number of experiments by two-thirds. Considering that the cooling effect of water primarily occurs at the center of the water surface and has a relatively small impact on the surrounding area [43], the interaction between this element and other factors is ignored. Only the interaction between PA, LSI, and PGA is considered. In Table 3, A, B, C, and D represent the factors PA, LSI, PGA, and PWA, respectively. A × B, A × C, and B × C denote the interactions between PA, LSI, and PGA. The levels of the factors are represented by 1, 2, and 3. Based on this, the layout of the 27 experimental scenarios (Table 4) was obtained. In these experimental scenarios, the water body was placed in the center and surrounded by plants, following common park layout patterns.

**Table 3.** Orthogonal table L27 ($3^{13}$).

| Experimental Scenario No. | A | B | A × B | A × B | C | A × C | A × C | B × C | D | Blank | B × C | Blank | Blank |
|---|---|---|---|---|---|---|---|---|---|---|---|---|---|
| 1 | 1 | 1 | 1 | 1 | 1 | 1 | 1 | 1 | 1 | 1 | 1 | 1 | 1 |
| 2 | 1 | 1 | 1 | 1 | 2 | 2 | 2 | 2 | 2 | 2 | 2 | 2 | 2 |
| 3 | 1 | 1 | 1 | 1 | 3 | 3 | 3 | 3 | 3 | 3 | 3 | 3 | 3 |
| 4 | 1 | 2 | 2 | 2 | 1 | 1 | 1 | 2 | 2 | 2 | 3 | 3 | 3 |
| 5 | 1 | 2 | 2 | 2 | 2 | 2 | 2 | 3 | 3 | 3 | 1 | 1 | 1 |
| 6 | 1 | 2 | 2 | 2 | 3 | 3 | 3 | 1 | 1 | 1 | 2 | 2 | 2 |
| 7 | 1 | 3 | 3 | 3 | 1 | 1 | 1 | 3 | 3 | 3 | 2 | 2 | 2 |
| 8 | 1 | 3 | 3 | 3 | 2 | 2 | 2 | 1 | 1 | 1 | 3 | 3 | 3 |
| 9 | 1 | 3 | 3 | 3 | 3 | 3 | 3 | 2 | 2 | 2 | 1 | 1 | 1 |
| 10 | 2 | 1 | 2 | 3 | 1 | 2 | 3 | 1 | 2 | 3 | 1 | 2 | 3 |
| 11 | 2 | 1 | 2 | 3 | 2 | 3 | 1 | 2 | 3 | 1 | 2 | 3 | 1 |
| 12 | 2 | 1 | 2 | 3 | 3 | 1 | 2 | 3 | 1 | 2 | 3 | 1 | 2 |
| 13 | 2 | 2 | 3 | 1 | 1 | 2 | 3 | 2 | 3 | 1 | 3 | 1 | 2 |
| 14 | 2 | 2 | 3 | 1 | 2 | 3 | 1 | 3 | 1 | 2 | 1 | 2 | 3 |
| 15 | 2 | 2 | 3 | 1 | 3 | 1 | 2 | 1 | 2 | 3 | 2 | 3 | 1 |
| 16 | 2 | 3 | 1 | 2 | 1 | 2 | 3 | 3 | 1 | 2 | 2 | 3 | 1 |
| 17 | 2 | 3 | 1 | 2 | 2 | 3 | 1 | 1 | 2 | 3 | 3 | 1 | 2 |
| 18 | 2 | 3 | 1 | 2 | 3 | 1 | 2 | 2 | 3 | 1 | 1 | 2 | 3 |
| 19 | 3 | 1 | 3 | 2 | 1 | 3 | 2 | 1 | 3 | 2 | 1 | 3 | 2 |
| 20 | 3 | 1 | 3 | 2 | 2 | 1 | 3 | 2 | 1 | 3 | 2 | 1 | 3 |
| 21 | 3 | 1 | 3 | 2 | 3 | 2 | 1 | 3 | 2 | 1 | 3 | 2 | 1 |
| 22 | 3 | 2 | 1 | 3 | 1 | 3 | 2 | 2 | 1 | 3 | 3 | 2 | 1 |
| 23 | 3 | 2 | 1 | 3 | 2 | 1 | 3 | 3 | 2 | 1 | 1 | 3 | 2 |
| 24 | 3 | 2 | 1 | 3 | 3 | 2 | 1 | 1 | 3 | 2 | 2 | 1 | 3 |
| 25 | 3 | 3 | 2 | 1 | 1 | 3 | 2 | 3 | 2 | 1 | 2 | 1 | 3 |
| 26 | 3 | 3 | 2 | 1 | 2 | 1 | 3 | 1 | 3 | 2 | 3 | 2 | 1 |
| 27 | 3 | 3 | 2 | 1 | 3 | 2 | 1 | 2 | 1 | 3 | 1 | 3 | 2 |

Note: A, B, C, and D represent PA, LSI, PGA, and PWA, respectively. A × B, A × C, and B × C represent the interactions between PA, LSI, and PGA. Blank columns indicate error columns. The numbers 1, 2, and 3 represent the levels of the factors.

Previous research has highlighted variations in park cooling effects on surrounding buildings of different heights [38]. To compare parks' impacts on high-rise (54 m) and low-rise (18 m) neighborhoods, two sets of orthogonal experiments were conducted, totaling 54 experimental scenarios, with 27 scenarios each for the low-rise and high-rise neighborhoods. Additionally, the study focused on a 300 m radius around the park, where green spaces exhibit pronounced cooling effects [41]. Meanwhile, the standard building spacing in Zhengzhou City (20 m for east–west direction, 40 m for south–north direction) was considered for the building layout of the blocks surrounding the park. Figure 2 showcases the 3D scene of the park and surrounding blocks, using experimental scenario 1 from low-rise (18 m) neighborhoods as an example.

**Table 4.** Layout of 27 experimental scenarios. Scenarios 1–9 have a park area of 5 ha, Scenarios 10–18 have a park area of 10 ha, and Scenarios 19–27 have a park area of 15 ha.

| Scenario 1 | Scenario 2 | Scenario 3 | Scenario 4 | Scenario 5 | Scenario 6 | Scenario 7 | Scenario 8 | Scenario 9 |
|---|---|---|---|---|---|---|---|---|
|  |  |  |  |  |  |  |  |  |
| A1B1C1D1 | A1B1C2D2 | A1B1C3D3 | A1B2C1D2 | A1B2C2D3 | A1B2C3D1 | A1B3C1D3 | A1B3C2D1 | A1B3C3D2 |
| PA:5ha LSI:1.13 PGA:60% PWA:0% | PA:5ha LSI:1.13 PGA:70% PWA:10% | PA:5ha LSI:1.13 PGA:80% PWA:20% | PA:5ha LSI:1.2 PGA:60% PWA:10% | PA:5ha LSI:1.2 PGA:70% PWA:20% | PA:5ha LSI:1.2 PGA:80% PWA:0% | PA:5ha LSI:1.4 PGA:60% PWA:20% | PA:5ha LSI:1.4 PGA:70% PWA:0% | PA:5ha LSI:1.4 PGA:80% PWA:10% |
| **Scenario 10** | **Scenario 11** | **Scenario 12** | **Scenario 13** | **Scenario 14** | **Scenario 15** | **Scenario 17** | **Scenario 17** | **Scenario 18** |
|  |  |  |  |  |  |  |  |  |
| A2B1C1D2 | A2B1C2D3 | A2B1C3D1 | A2B2C1D3 | A2B2C2D1 | A2B2C3D2 | A2B3C1D1 | A2B3C2D2 | A2B3C3D3 |
| PA:10ha LSI:1.13 PGA:60% PWA:10% | PA:10ha LSI:1.13 PGA:70% PWA:20% | PA:10ha LSI:1.13 PGA:80% PWA:0% | PA:10ha LSI:1.2 PGA:60% PWA:20% | PA:10ha LSI:1.2 PGA:70% PWA:0% | PA:10ha LSI:1.2 PGA:80% PWA:10% | PA:10ha LSI:1.4 PGA:60% PWA:0% | PA:10ha LSI:1.4 PGA:70% PWA:10% | PA:10ha LSI:1.4 PGA:80% PWA:20% |
| **Scenario 19** | **Scenario 20** | **Scenario 21** | **Scenario 22** | **Scenario 23** | **Scenario 24** | **Scenario 25** | **Scenario 26** | **Scenario 27** |
|  |  |  |  |  |  |  |  |  |
| A3B1C1D3 | A3B1C2D1 | A3B1C3D2 | A3B2C1D1 | A3B2C2D2 | A3B2C3D3 | A3B3C1D2 | A3B3C2D3 | A3B3C3D1 |
| PA:15ha LSI:1.13 PGA:60% PWA:20% | PA:15ha LSI:1.13 PGA:70% PWA:0% | PA:15ha LSI:1.13 PGA:80% PWA:10% | PA:15ha LSI:1.2 PGA:60% PWA:0% | PA:15ha LSI:1.2 PGA:70% PWA:10% | PA:15ha LSI:1.2 PGA:80% PWA:20% | PA:15ha LSI:1.4 PGA:60% PWA:10% | PA:15ha LSI:1.4 PGA70% PWA:20% | PA:15ha LSI:1.4 PGA:80% PWA:0% |

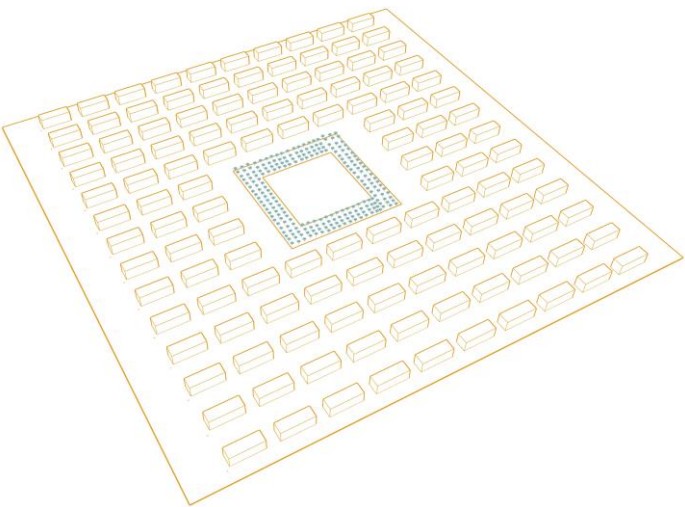

**Figure 2.** Three-dimensional scene of the park and surrounding blocks, using experimental scenario 1 from low-rise (18 m) neighborhoods as an example.

2.2.2. Experimental Design Analysis

Considering residents' activities during the morning (6:00–8:00) and evening (18:00–20:00), the period from 13:00 to 15:00 has relatively adverse microclimatic conditions. Studies show that greenery has a greater cooling effect during this time. Thus, for analysis, we will focus on 7:00, 14:00, and 19:00 as representative time points.

The range analysis method was used to determine the primary and secondary influences of each factor and their optimal levels. ANOVA was then employed to quantitatively analyze the results, providing insights into the significance and specific impact of factors and their interactions on the temperature difference inside and outside the park. ANOVA was also used to calculate the contribution rates of the influencing factors. Based on the orthogonal experiment and variance analysis, the contribution rate of each factor ($\rho_j$/%) was computed using the sum of squared deviations (Formulas (2) and (3)) divided by the total sum of squared deviations (Formulas (4) and (5)).

$$s_j = \frac{b}{a}\sum_{k=1}^{b}\left(\overline{y_{ik}} - \overline{y}\right)^2 = \frac{b}{a}\sum_{k=1}^{b} y_{jk}^2 - \frac{1}{a}\left(\sum_{i=1}^{a} y_i\right)^2 \tag{2}$$

$$\sum_{k=1}^{b} y_{jk}^2 = K_1^2 + K_2^2 + K_3^2 \tag{3}$$

$$S = \sum_{i=1}^{a}\left(y_i - \overline{y}\right)^2 = \sum_{i=1}^{a} y_i^2 - \frac{1}{a}\left(\sum_{i}^{a} y_i\right)^2 \tag{4}$$

$$\rho_j = \frac{S_j - \frac{s_e}{f_e}f_j}{S} \times 100\% \tag{5}$$

In the formula, $S_j$ represents the column's squared deviations; $S$ represents the total squared deviations; $a$ denotes the number of experiments; $b$ denotes the number of levels; $j$ represents the column number; $i$ represents the experiment number; $y_i$ represents the target performance; $K_1$, $K_2$, and $K_3$ represent the summation values of the 1st, 2nd, and 3rd levels in the same column; $s_e$ represents the squared errors; $f_j$ represents the degrees of freedom for the influencing factors; and $f_e$ represents the variance of the errors.

Furthermore, for the factors that have significant interactions confirmed through ANOVA, the average temperature difference (external minus internal) was calculated for all combinations of levels of these two factors. This represents the interaction between these two factors at that specific level. For example, in the experiment, there are three scenarios that include the combination of A1 (PA 5 ha) and B1 (LSI 1.13), namely Scenario 1, Scenario

2, and Scenario 3. The average temperature difference between the interior and exterior of the park in these three scenarios represents the interaction effect between A1 and B1.

Taking into account that the impact of landscape design parameters on the cooling effect of parks varies with time, it is necessary to consider the weighting of time when optimizing the design scheme. Urban park cooling effects are stronger during the day compared to the night, as seen in previous studies [44,45]. The influence weights for the three time points (7:00, 14:00, 19:00) were set at 20%, 60%, and 20% respectively. By weighting the time differences, the overall cooling impact of the park on the surrounding area was assessed.

### 2.3. Envi-Met Simulation

2.3.1. Software Introduction

ENVI-met (version 5.0.3), a German-developed software, was selected for this study. It is a 3D urban microclimate simulation tool that models interactions among buildings, vegetation, ground surfaces, and the atmosphere at a high-resolution scale (0.5 m–10 m). It calculates simulation results for wind, temperature, moisture, and solar radiation at the microclimate level [32–34]. ENVI-met (version 5.0.3) is widely used in urban environmental design.

2.3.2. Software Verification

Zijingshan Park in Zhengzhou, the city's oldest and largest urban park, was chosen for on-site measurements (Figure 3a). It spans about 800 m (east–west) and 370 m (north–south), offering diverse landscapes and activities. Five strategically positioned measurement points were established within the park, all of which are located near pedestrian walkways. These points represent spaces where park visitors frequently stay and where environmental differences exist. Among them, Measurement Point 1 is situated inside a pavilion on the mountaintop. Measurement Point 2 is located in a larger activity square with trees around it. Measurement Point 3 is in close proximity to water. Measurement Point 4 is located under a small tree in a relatively enclosed environment. Measurement Point 5 is located under a large tree, with an open surrounding area. Due to the summer peak of high temperatures in Zhengzhou usually occurring in July [38], the testing date was set for 11 July 2022, and testing lasted from 7:00 a.m. to 6:00 p.m. HOBO U23-001A temperature and humidity loggers were selected and placed at a height of 1.5 m (Figure 3d). Louvered radiation shields were used to minimize solar interference (Figure 3c). Table 5 provides details on the measurement instruments' range and accuracy. The park model (190 × 90 × 10) was constructed based on the current state of Zijingshan Park, with a 5 m grid resolution. Eight nested grids surrounded the park, and model parameters accurately represented the park's conditions. Simulated results used meteorological data collected at 1.5 m height. Figure 4 compares the simulated and measured values of Ta and RH at the five measurement points. We use the root mean square error (RMSE) to analyze the differences between measured and simulated values. Previous research has indicated that the acceptable RMSE for Ta is less than 4.30 °C, and the acceptable value for RH should not exceed 10.2% [34]. Additionally, we used Willmott's index of agreement d to evaluate the accuracy of the model, where a value closer to 1 indicates a smaller prediction error [46]. In this study, the RMSE values of Ta and RH were 1.65 °C and 3.91%, respectively, and the d values of Ta and RH were, 0.93 and 0.96 respectively. The simulation errors were within an acceptable range, indicating that the ENVI-met model constructed in this study is capable of accurately simulating the real environmental conditions of the Zijingshan Park.

**Table 5.** Field measurement instruments, ranges, and accuracy.

| Microclimatic Parameters | Instrument | Range | Accuracy |
| --- | --- | --- | --- |
| Air temperature (Ta) | HOBO U23-001A | −40–70 °C | ±0.21 °C |
| Relative humidity (RH) | HOBO U23-001A | 0–100% | ±2.5% |

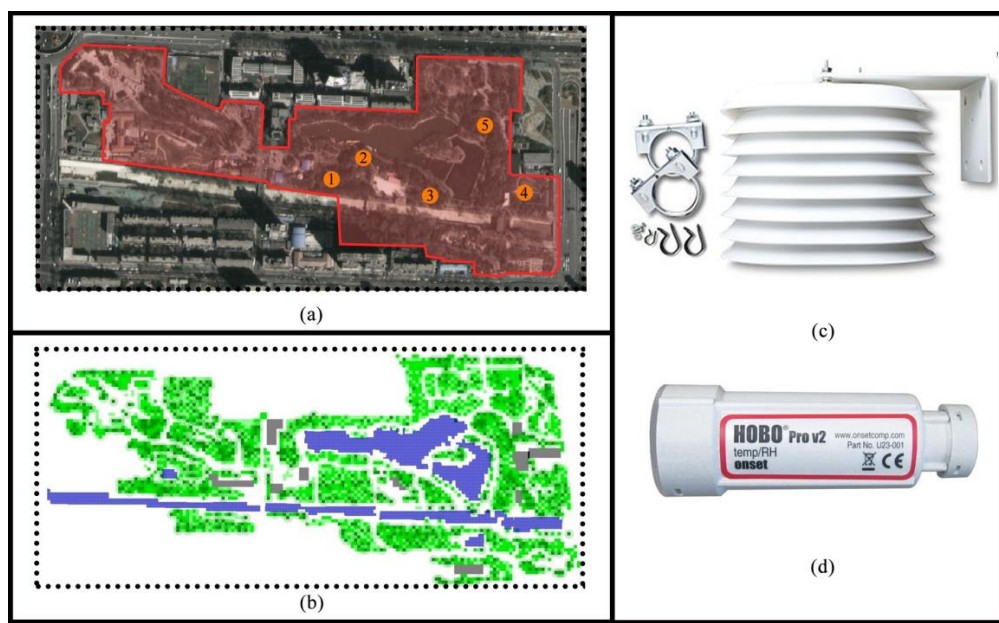

**Figure 3.** Ideal model and field measurement setup. (**a**) Satellite image and field measurement locations. The numbers represent the positions of the measurement points. (**b**) Ideal model. (**c**) Louvered radiation shield. (**d**) HOBO temperature and humidity data logger.

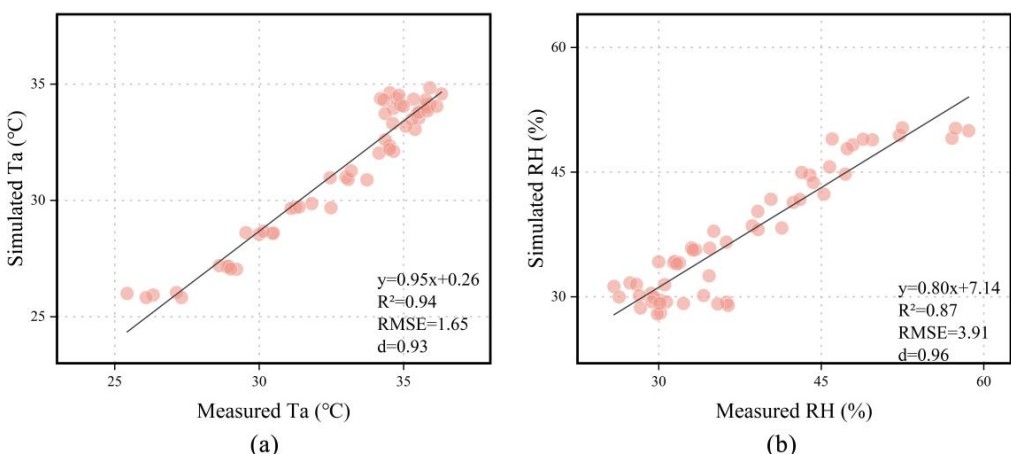

**Figure 4.** Ta and RH simulation and experimental fit. (**a**) Validation of Ta. (**b**) Validation of RH.

### 2.3.3. Simulation Environment Configuration

According to Section 2.2.1, we needed to simulate a total of 54 experimental scenario models for both low-rise neighborhoods and high-rise neighborhoods. The model domain consisted of multiple three-dimensional grid cells, with a horizontal resolution of 10 m (dx and dy) and a vertical resolution of 6 m (dz) for each individual grid cell. The number of grids for each model varied depending on the changes in the park area in the experimental scenario. Due to the simulated data being collected at a height of 1.5 m, a vertical grid with 5% telescoping from 2 m height was set up to accommodate tall buildings in the blocks and save simulation time [42]. In addition, considering the boundary effects caused by the software, 5 nested grids were set up as buffer zones in the Envi-met model to alleviate the instability when simulating elements near the border of the main model domain. Simultaneously, it was ensured that the model height was at least twice the height of the buildings. Additionally, ENV-met simulations need some information about the surrounding meteorology. In this study, simple forcing of meteorological boundary conditions was used to initialize the simulation. The input meteorological parameters were

obtained from typical summer day data in Zhengzhou. Each scenario used the same input meteorological data to avoid the influence of background climate. Each model was run for 24 h from 0:00 to 23:00 on 11 July 2022. Please refer to Table 6 for specific parameters.

**Table 6.** ENVI-met simulation parameter settings.

| Parameter Type | Relevant Parameters | Initial Data |
|---|---|---|
| Simulation Time | Simulation Latitude–Longitude | 34° N, 112° W |
| | Start Date and Time | 11 July 2022 |
| | Start Time | 0:00 |
| | Duration | 24 h |
| | Building Deformation Height | 2 m, 5% |
| | Time Step | 5 s |
| | Initial Ta | 25.60 °C |
| Initial Meteorological Parameters | Initial RH | 84% |
| | Wind Speed at 10 m | 3 m/s |
| | Wind Direction | 135° |
| | Ground Roughness | 0.01 |
| | Model Orientation | North–South |
| | Greenery | Trees (T1) |
| Model Materials | Impervious Surface | Underlayment (Loamy Soil) Pavement (Dark Granite) |
| | Water | Deep Water |

## 3. Results

### 3.1. Comparison and Analysis of Temperature Differences Inside and Outside the Park in Experimental Scenarios

We extracted average Ta values at 7:00, 14:00, and 19:00 for park grid cells. Meanwhile, average air temperature values were calculated for nearby grid cells (within 300 m) outside the park. The difference between the average air temperature outside and inside the park (outside minus inside) is used to represent the park's influence on the surrounding thermal environment. If the temperature difference is greater than 0, it indicates that the park has a cooling effect. If the temperature difference is less than 0, it indicates that the park has a warming effect.

#### 3.1.1. Characteristics of Temperature Differences Inside and Outside the Park in Different Scenarios

As shown in Figure 5, at 18 m building height, the park shows a cooling effect compared to the surrounding building blocks at 14:00. However, at 7:00 and 19:00, the park areas experience a slight temperature increase due to weaker solar radiation, lower plant transpiration, and slow heat dissipation in under-tree space. At 14:00, Scenario 3 has the maximum temperature difference of 0.67 °C with PA 5 ha, LSI 1.13, PGA 80%, and PWA 20%. Scenario 22 has the minimum temperature difference of 0.26 °C with PA 15 ha, LSI 1.2, PGA 60%, and PWA 0%. At 54 m building height at 14:00, compared to the 18 m building block scenario, a more pronounced temperature reduction is observed in Figure 5. At 14:00, Scenario 18 shows the maximum temperature difference of 0.89 °C with PA 10 ha, LSI 1.4, PGA 80%, and PWA 20%. Scenario 3 has the minimum temperature difference of 0.39 °C with PA 5 ha, LSI 1.13, PGA 80%, and PWA 20%. These two scenes have the same PWA and PGA, but the PA and LSI are different, which indicates that in high-rise neighborhoods, the size and shape of the park may have a significant impact on its cooling effect.

#### 3.1.2. Comparison of Noon Temperature Reduction between Two Building Heights

Figure 6 presents the temperature variations of the neighborhood within a 300 m radius surrounding the park, in which the height of the buildings is 18 m, in Scenario 3 (max difference) and Scenario 22 (min difference). The average temperature in the park in Scenario 3 is close to that in Scenario 22. As shown in Figure 6a, the westward temperature ranges span from 31.43 °C to 32.28 °C with a difference of 0.85 °C in Scenario 3 and from 31.33 °C to 31.96 °C with a difference of 0.63 °C in Scenario 22. Figure 6b illustrates the

eastward range, spanning from 32.48 °C to 33.09 °C with a difference of 0.61 °C in Scenario 3 and from 32.26 °C to 32.85 °C with a difference of 0.57 °C in Scenario 22. Figure 6c demonstrates the northward range, spanning from 31.90 °C to 32.72 °C with a difference of 0.82 °C in Scenario 3 and from 31.75 °C to 32.11 °C with a difference of 0.36 °C in Scenario 22. Finally, Figure 6d displays the southward range, spanning from 32.00 °C to 33.19 °C with a difference of 1.19 °C in Scenario 3 and from 31.61 °C to 31.86 °C with a difference of 0.25 °C in Scenario 22.

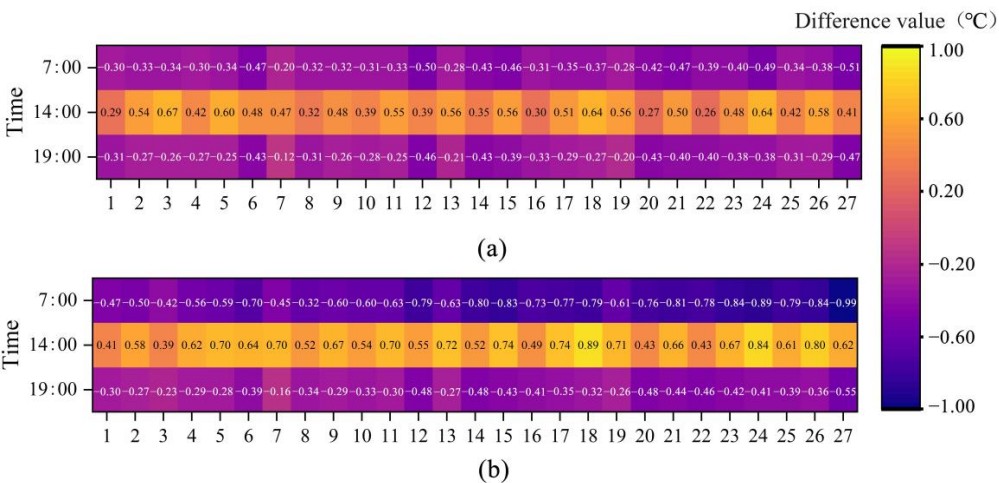

**Figure 5.** Temperature differences inside and outside park for 54 scenarios: (**a**) 18 m building height; (**b**) 54 m building height.

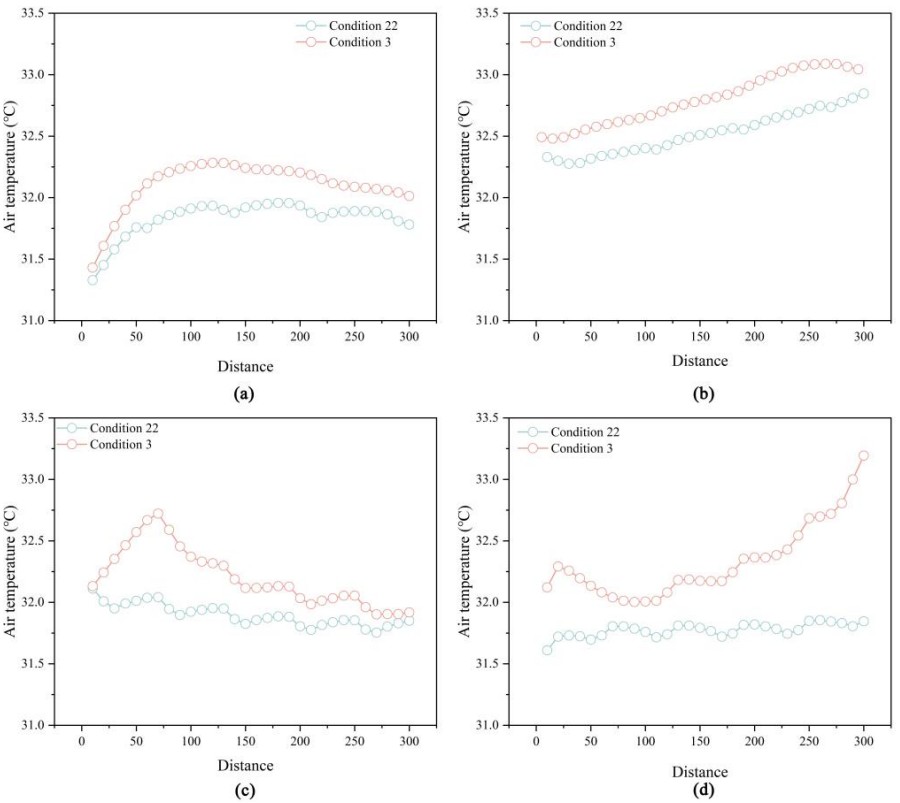

**Figure 6.** At 14:00, parks in Scenario 3 (max difference) and Scenario 22 (min difference), surrounded by 18 m high buildings, exhibit cooling effects on the surrounding area. The impacts of Scenario 3 and Scenario 22 are showcased in four directions: (**a**) westward (300 m); (**b**) eastward (300 m); (**c**) northward (300 m); (**d**) southward (300 m).

The park's cooling effects on the neighborhood in four directions in Scenario 22 are all greater than those in Scenario 3, especially in the south direction. In addition, the cooling effect of the park varies in different directions as well. In Scenario 22, the park exhibits a stable cooling effect in the north and south directions, with minimal temperature fluctuations within a 300 m range on both sides. The air temperature on the south side of the park remains almost constant, while on the eastern side, the air temperature continues to rise. Due to the sun being positioned in the southwest direction at 14:00, the buildings on the western side of the park can block some of the solar radiation, creating shadows. This results in the air temperature within a 60 m range on the western side of the park being lower than the average temperature inside the park. In Scenario 3, the air temperature fluctuations are relatively large in four directions, but there are some differences in the characteristics of the fluctuations. Similar to Scenario 22, the air temperature continues to rise on the eastern side of the park. In the other three directions, the air temperature undergoes a significant change near the vicinity of 70 m outside the park, but the change trends are different. After surpassing 70 m on the western side of the park, the air temperature starts to relatively stabilize, while on the northern side, the air temperature experiences a sudden decrease, and on the southern side, it is the opposite.

Figure 7 shows the temperature variations of the neighborhood within a 300 m radius surrounding the park, in which the height of the buildings is 54 m, in Scenario 18 (max difference) and Scenario 3 (min difference). The park's average temperature is 31.21 °C in Scenario 18 and 31.85 °C in Scenario 3. In Figure 7a, the westward temperature ranges span from 31.08 °C to 31.96 °C with a difference of 0.88 °C in Scenario 18 and from 31.30 °C to 32.27 °C with a difference of 0.97 °C in Scenario 3. Figure 7b illustrates the eastward range, spanning from 32.25 °C to 32.86 °C with a difference of 0.61 °C in Scenario 18 and from 32.36 °C to 32.93 °C with a difference of 0.57 °C in Scenario 3. Figure 7c demonstrates the northward range, spanning from 31.99 °C to 32.55 °C with a difference of 0.56 °C in Scenario 18 and from 31.78 °C to 32.57 °C with a difference of 0.79 °C in Scenario 3. Finally, Figure 7d displays the southward range, spanning from 31.66 °C to 32.28 °C with a difference of 0.62 °C in Scenario 18 and from 31.89 °C to 32.95 °C with a difference of 1.06 °C in Scenario 3. Overall, the cooling effect of Scenario 18 is superior to that of Scenario 3. The temperature variation characteristics of the eastern and western sides of the park in the high-rise neighborhood during midday are generally consistent with those in the low-rise neighborhood. However, influenced by the solar azimuth angle, the shadows of high-rise buildings mainly fall on the northern side of the buildings at noon, resulting in frequent temperature fluctuations within a certain range in the southern and northern areas outside the park.

### 3.2. The Impact of Landscape Design Parameters on Park Temperature Difference

Temperature reduction, measured as the difference between zone temperatures and the park, evaluates parks' impact on nearby blocks. This study explores factors influencing temperature reduction. Analyzing these factors enables weighted plan comparisons.

### 3.2.1. Ranking of Primary and Secondary Impacts and Optimal Factor Levels

The range analysis method was used to analyze orthogonal experiment results. Average temperature differences between the park's interior and exterior at different factor levels were calculated, yielding factor ranges. At 18 m building height, temperature difference trends between levels and the park's interior/exterior were shown at 14:00 and 19:00 (Figures 8 and 9). The most significant difference was observed at 14:00, reaching a maximum of 0.59 °C, followed by 19:00, while the smallest difference occurred at 7:00. Among the factors, PWA had the largest range of 0.24 at 14:00, with PGA and LSI having minor effects. Similar temperature difference trends were seen at 54 m building height compared to the 18 m scenario (Figures 8 and 9). The most pronounced difference occurred at 14:00, reaching a maximum of 0.72 °C, while the differences at 7:00 and 19:00 were

smaller. PA had the greatest influence on temperature difference, with a maximum range of 0.27 °C, followed by PWA, and PGA and LSI had minor effects.

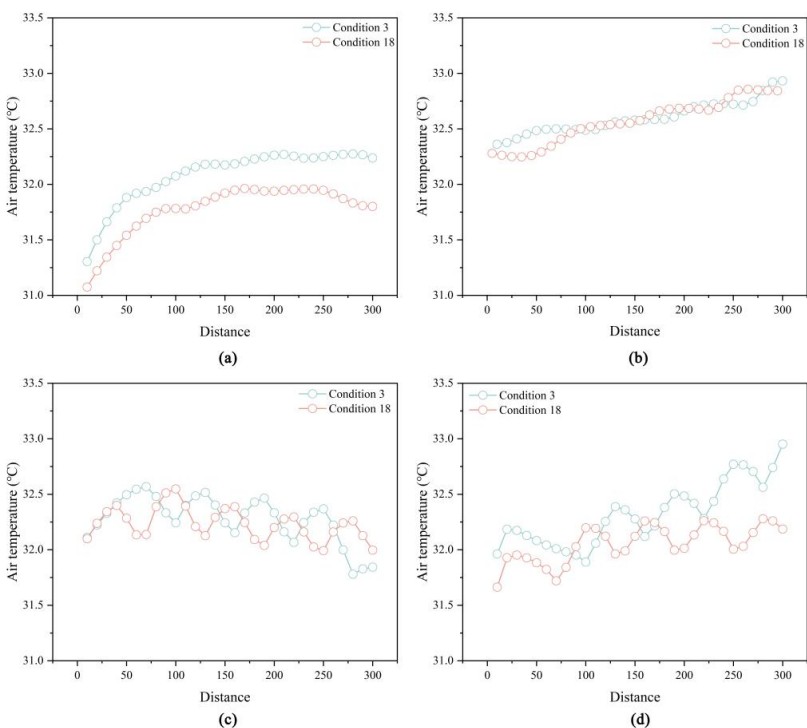

**Figure 7.** At 14:00, parks in Scenario 18 (max difference) and Scenario 3 (min difference), surrounded by 54 m high buildings, exhibit cooling effects on the surrounding area. The impacts of Scenario 18 and Scenario 3 are showcased in four directions: (**a**) westward (300 m); (**b**) eastward (300 m); (**c**) northward (300 m); (**d**) southward (300 m).

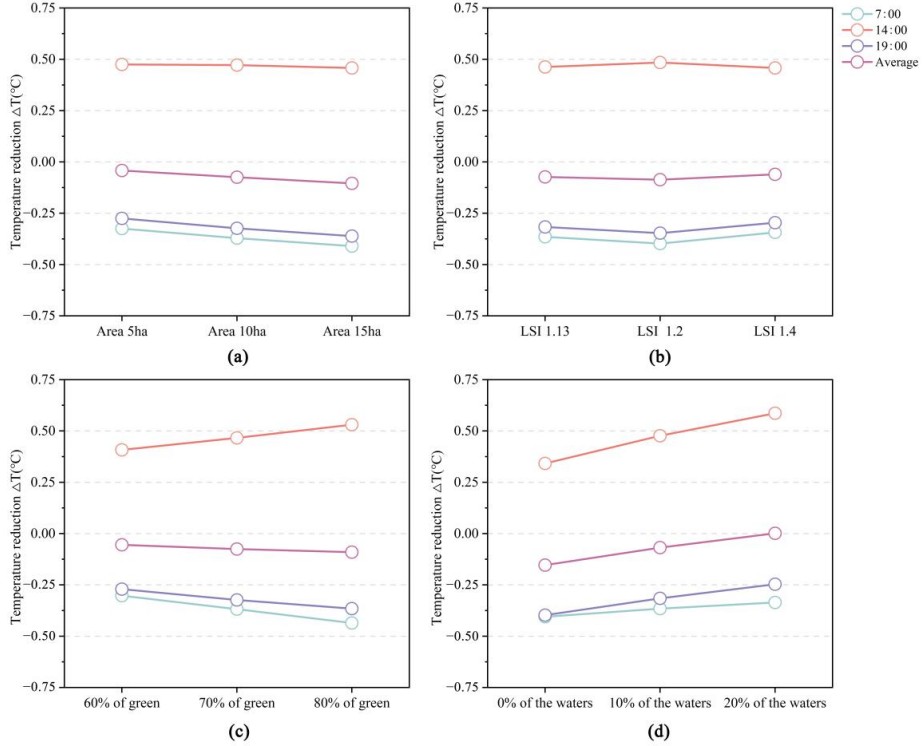

**Figure 8.** Building height 18 m—impact of different factor levels in building blocks on temperature differences between the park's exterior and interior: (**a**) PA; (**b**) LSI; (**c**) PGA; (**d**) PW.

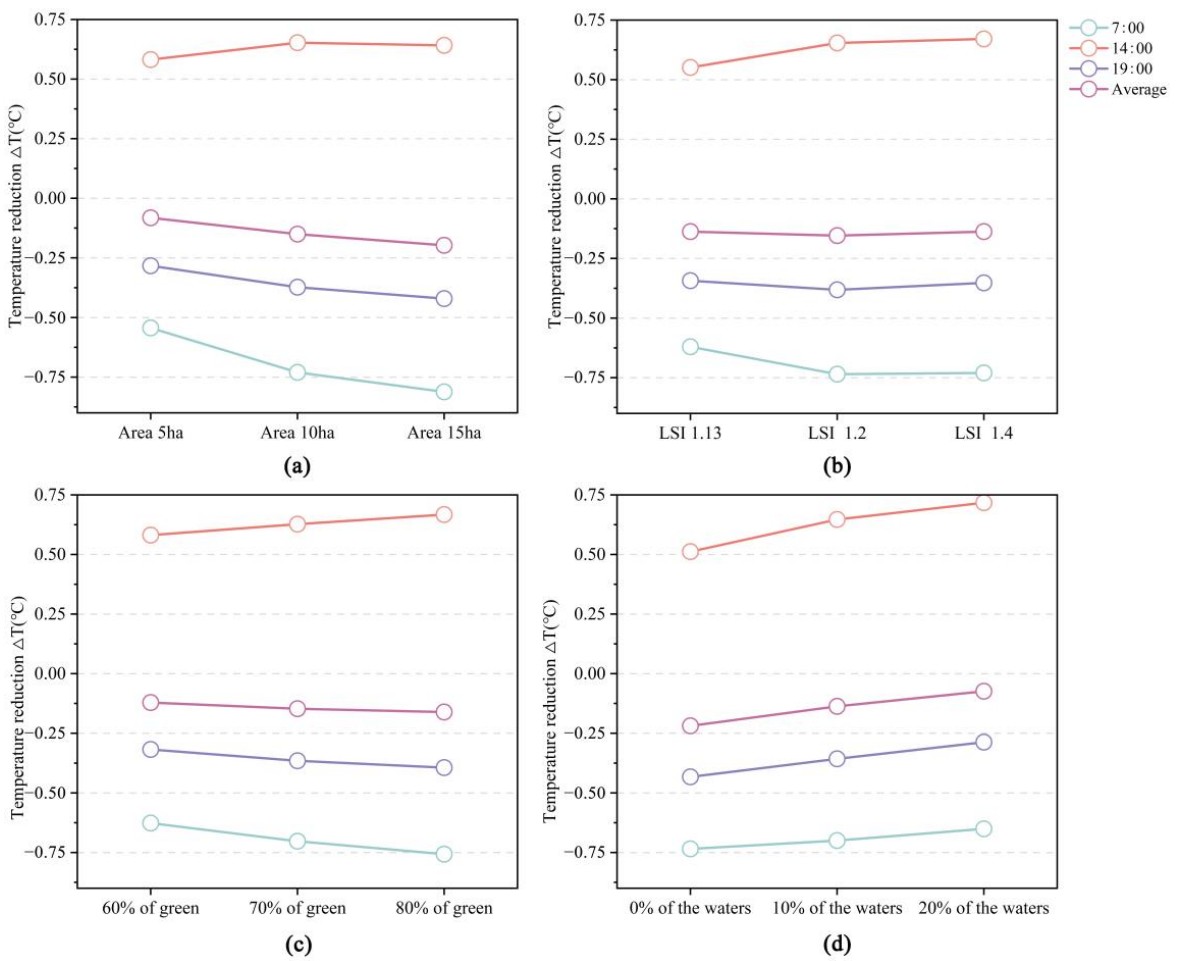

**Figure 9.** Building height 54 m—impact of different factor levels in building blocks on temperature differences between the park's exterior and interior: (**a**) PA; (**b**) LSI; (**c**) PGA; (**d**) PWA.

Comparing temperature differences inside and outside the park in low-rise and high-rise building blocks, the 54 m scenario consistently shows larger differences than the 18 m scenario. This indicates the significant impact of adjusting park design parameters on the surrounding high-rise residential areas. Compared to low-rise neighborhoods, each landscape design factor in high-rise neighborhoods contributes to a cooling increase of 0.27 °C to 0.64 °C at noon.

### 3.2.2. Factor Influence and Significance Analysis

Factor significance analysis (Table 7): For 18 m building height, PA, LSI, PGA, and PWA significantly impact temperature difference inside and outside the park during morning and evening ($p < 0.05$). An analysis of F-values (Table 7) and contribution rates (Figure 10) reveals the following factor influence order at 7:00: PGA (50.43%) > PA (20.65%) > PWA (13.88%) > LSI (8.48%). At 19:00, the order is as follows: PWA (52.30%) > PGA (20.87%) > PA (16.98%) > LSI (6.05%). At 14:00, only PGA and PWA have a significant impact, ranking PWA (73.02%) > PGA (18.24%). Other factors' contribution rates were <1% and negligible. For 54 m building height, PA, LSI, PGA, and PWA significantly impact temperature difference during morning and evening, consistent with the results for 18 m blocks. Referring to F-values (Table 7) and contribution rates (Figure 10), the order at 7:00 is as follows: PA (62.11%) > PGA (14.05%) > LSI (13.84%) > PWA (5.89%). At 19:00, the order is as follows: PWA (42.41%) > PA (39.36%) > PGA (11.79%) > LSI (3.23%). At 14:00, only LSI and PWA have a significant impact, ranking PWA (43.99%) > LSI (16.90%). Other factors' influence was relatively small and negligible.

**Table 7.** F-values and significance of variance analysis for three time periods in different building height blocks, with significance levels of $p < 0.05$ * and $p < 0.01$ **.

| Building Height | 18 m | | | 54 m | | |
|---|---|---|---|---|---|---|
| Time | 7:00 | 14:00 | 19:00 | 7:00 | 14:00 | 19:00 |
| A (PA) | 114.18 ** | 0.55 | 120.58 ** | 662.68 ** | 1.81 | 164.10 ** |
| B (LSI) | 46.92 ** | 1.37 | 42.95 ** | 147.66 ** | 5.22 * | 13.46 ** |
| C (PGA) | 278.90 ** | 25.72 ** | 148.23 ** | 149.95 ** | 2.31 | 49.15 ** |
| D (PWA) | 76.76 ** | 102.96 ** | 371.43 ** | 62.82 ** | 13.59 ** | 176.8 ** |
| A * B | 10.70 ** | 2.63 | 8.69 * | 11.68 ** | 0.67 | 3.31 |
| A * C | 3.58 | 0.62 | 1.62 | 7.73 * | 0.79 | 1.24 |
| B * C | 2.36 | 0.44 | 1.66 | 0.99 | 1.02 | 0.64 |

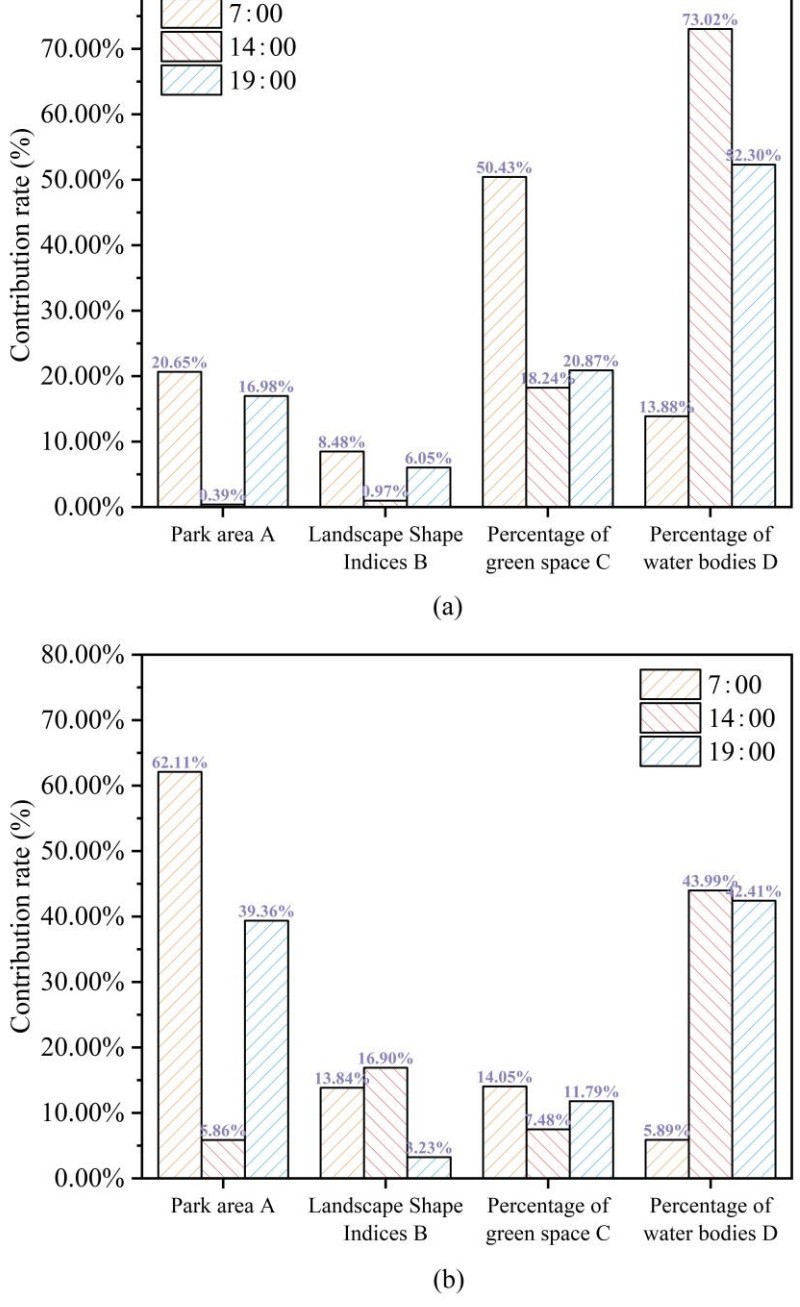

**Figure 10.** Contribution rates of factors to park temperature difference: (**a**) building height: 18 m; (**b**) building height: 54 m.

As shown in Figure 10a (18 m), PA, LSI, and PGA have varying contributions to temperature differences inside and outside the park over time. PA, LSI, and PGA show a decreasing-then-increasing trend, while PWA shows the opposite trend. PGA has the highest morning contribution, PWA dominates at noon and evening, and LSI consistently has the lowest contribution rate. As shown in Figure 10b (54 m), PA and PGA follow a similar decreasing-then-increasing trend, and LSI and PWA exhibit the opposite trend. PA has the highest morning contribution, PWA dominates at noon, and both factors play a crucial role in the evening, with combined contribution rates exceeding 80%.

### 3.3. Optimal Combination of Factor Interactions

To identify optimal factor interactions and their combinations, we calculated average temperature differences between the park exterior and interior (exterior minus interior) during different time periods and plotted an interaction chart (Figure 11) for the factors that exhibited significant interaction effects. Each cell represents the mean effect of a specific factor combination, e.g., A1B1: PA 5 ha and LSI 1.13 combo average.

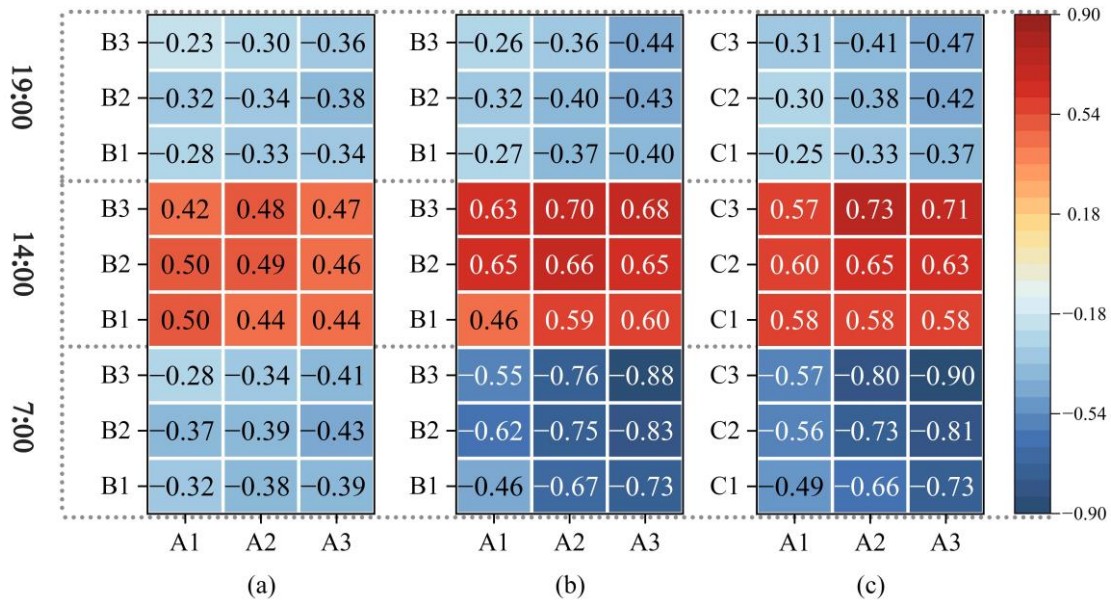

**Figure 11.** Interactions of factors in building blocks: (**a**) 18 m height—A and B; (**b**) 54 m height—A and B; (**c**) 54 m height—A and C. The values represent the average temperature difference between the exterior and interior of the park (exterior minus interior). A value greater than 0 (shown in red) indicates a temperature decrease, while a value less than 0 (shown in blue) indicates a temperature increase.

For the 18 m building scenario, Figure 11 shows PA (A) and LSI (B) interaction effects. At 14:00, A1B1 and A1B2 had the highest cooling, averaging 0.50 °C. A1B3 performed poorly, indicating the unsuitability of the elongated rectangular park shape for small areas, reducing cooling. A2B2, A2B3, and A3B3 also showed significant cooling. Larger parks benefit from elongated rectangular or irregular shapes, enhancing airflow exchange and improving the nearby thermal environment. A1Bi (i = 1, 2, 3) performed best in the morning and evening, minimizing warming.

For the 54 m building scenario, Figure 11 depicts the interaction effects of PA (A) and LSI (B), as well as PA (A) and PGA (C). At 7:00, AiBi and AiCi (i = 1, 2, 3) combinations cause significant warming, especially A3C3 with 0.90 °C higher temperature than the overall area. At 14:00, A1B1 has the lowest cooling effect (0.46 °C for A1B1), unlike the 18 m case. However, A2B3 and A3B3 combinations exhibit substantial cooling. Thus, a large area with an elongated rectangular park layout is recommended for better cooling near a 54 m building, resulting in a 0.70 °C reduction. For LSI 1.2 (B2), different PA values yield similar cooling effects. Among A and C interactions at 14:00, A2C3 shows maximum

cooling (0.73 °C). Comparing A2Ci and A3Ci (i = 1, 2, 3), A2Ci generally demonstrates better cooling within the same PGA conditions, indicating that higher PA values do not guarantee greater cooling.

### 3.4. Optimized Experimental Design

Landscape design parameters have varying impacts on the temperature difference between the park and its surroundings, and these impacts change over time. Therefore, it is necessary to weight the time differences to more accurately assess the overall cooling impact of the park on the surrounding areas. The results of 54 experimental scenarios for two building height districts were weighted and averaged according to the time weighting mentioned in Section 2.2.2. PA-based categorization and ranking of comprehensive cooling effectiveness are shown in Table 8. In the 54 m building height district, weighted cooling amplitudes were higher than those in the 18 m building height district, aligning with previous findings on primary and secondary factors. For 18 m height, optimal combinations were as follows: PA 5 ha—Scenario 3: LSI 1.13, PGA 80%, PWA 20%; PA 10 ha—Scenario 18: LSI 1.4, PGA 80%, PWA 20%; PA 15 ha—Scenario 19: LSI 1.13, PGA 60%, PWA 20%. In the 54 m height district, the optimal combination for PA 5 ha was Scenario 7: LSI 1.4, PGA 60%, PWA 20%. PA 10 ha and 15 ha had the same optimal combinations as the 18 m building height district.

**Table 8.** Weighted results.

| Building Height | 5 ha | | 10 ha | | 15 ha | |
|---|---|---|---|---|---|---|
| | Experiment No. | Weighted Values (°C) | Experiment No. | Weighted Values (°C) | Experiment No. | Weighted Values (°C) |
| 18 m | 1 | 0.05 | 10 | 0.11 | 19 | 0.24 |
| | 2 | 0.20 | 11 | 0.21 | 20 | −0.01 |
| | 3 | 0.28 | 12 | 0.04 | 21 | 0.13 |
| | 4 | 0.14 | 13 | 0.24 | 22 | 0.00 |
| | 5 | 0.24 | 14 | 0.04 | 23 | 0.13 |
| | 6 | 0.11 | 15 | 0.17 | 24 | 0.21 |
| | 7 | 0.22 | 16 | 0.05 | 25 | 0.12 |
| | 8 | 0.07 | 17 | 0.18 | 26 | 0.21 |
| | 9 | 0.17 | 18 | 0.26 | 27 | 0.05 |
| 54 m | 1 | 0.09 | 10 | 0.14 | 19 | 0.25 |
| | 2 | 0.19 | 11 | 0.23 | 20 | 0.01 |
| | 3 | 0.11 | 12 | 0.08 | 21 | 0.15 |
| | 4 | 0.20 | 13 | 0.25 | 22 | 0.01 |
| | 5 | 0.24 | 14 | 0.06 | 23 | 0.15 |
| | 6 | 0.17 | 15 | 0.19 | 24 | 0.25 |
| | 7 | 0.30 | 16 | 0.06 | 25 | 0.13 |
| | 8 | 0.12 | 17 | 0.22 | 26 | 0.24 |
| | 9 | 0.22 | 18 | 0.31 | 27 | 0.07 |

## 4. Discussion

### 4.1. Main Findings

As an important component of urban forests, parks have been proven to play a significant and positive role in mitigating the urban heat island effect [6,47]. Identifying landscape design parameters significantly correlated with temperature in urban parks under different climatic conditions and exploring urban green space planning to enhance cooling potential have become frontier topics in urban thermal environment research [47–49]. The cooling potential of a park is influenced by many landscape design parameters, such as the size, shape, vegetation coverage, and water surface area of the park [47,50–52]. Although some studies have attempted to clarify the relationship between a park's landscape design parameters and its cooling capacity, the comprehensive impact of landscape design parameters on a park's cooling capacity under synergistic effects has not been accurately determined

yet. Previous studies have indicated that increasing park area and improving the ratio of blue and green spaces can effectively enhance a park's cooling capacity [53]. Many studies analyze the surface temperature derived from remote sensing imagery at specific moments and suggest that a park's area is the primary influencing factor for its cooling capacity [29,50,54]. The present study, based on continuous-time simulation, found that the contributions of landscape design parameters to the park's cooling effect vary over time. Park area primarily influences the cooling capacity of the park during the morning and evening. In addition, the cooling effect of the park is not only influenced by internal landscape elements, but also related to spatial morphological indicators of the surrounding urban neighborhood, such as building density, building height, and sky view factor [55–57]. Therefore, this study attempts to compare two typical urban blocks with different building heights to explore the cooling effect of parks under different combinations of landscape design parameters. The results indicate that parks have a more pronounced cooling effect on the surrounding urban area during midday. However, the cooling efficiency varies depending on the direction. In these two typical urban blocks with different building heights, the park's cooling capacity on the southern block is most influenced by landscape design parameters, while that on the eastern block is less affected. In comparison to the low-rise building block, the park's cooling effect on the high-rise building block is more significant, which may be because high-rise buildings can create a barrier that prevents the dense air near the green space from flowing out, thus maintaining the cooling effect of the park [58]. In low-rise blocks, the PGA and PWA have a significant influence on the cooling effect of the park during midday, while the impact of the LSI and PWA is significant in high-rise blocks. This may be due to the impact of canyon winds which is more likely to appear in high-rise blocks, and in these four landscape design parameters, the shape and area of the park are evidently more closely related to the wind environment of the surrounding urban space. Furthermore, taller buildings tend to cast larger shadows. If the park has a small area, the green spaces are more likely to be within the shadow of the buildings. It is discovered that the cooling effect of the trees located in the shadow of the buildings is weaker than that of the trees exposed to solar radiation [59].

### 4.2. Planning Recommendation

Through rational planning of PA, LSI, and PWA, the summer thermal environment of surrounding neighborhoods can be improved. For 18 m buildings, an LSI below 1.2 enhances park cooling, and a uniform park boundary promotes better thermal exchange. A larger PA reduces overall neighborhood temperatures, considering layout requirements. A PA around 10 ha suits typical urban blocks. In dense city centers, consider the main landscape factors. Optimize the layout for increased cooling. Avoid excessive densities in greenery to maintain natural ventilation and extend park cooling to the surroundings. Optimize the PGA for better nearby radiation. Note temperature increases during mornings and evenings. A lower LSI (<1.2) for 18 m buildings improves park cooling. Rectangular parks have greater cooling with equal PGA and PWA. Overlapping shadows provide moderate cooling. Tall buildings create airflow channels. Large parks significantly enhance cooling (PGA 60%–70%), without excessive density. Maintain 20% PWA for parks with different heights to lower neighborhood Ta, ensuring a balanced layout.

### 4.3. Limitations and Further Directions

This study applied the ENVI-met model to simulate urban park microclimates under various combinations of landscape design parameters. Some limitations should be addressed to help understand the results. First, the present study explored the effect of the park on the microclimate of the surrounding neighborhood during the summer season. In the future, it would be beneficial to expand the study to four seasons and comprehensively evaluate the climatic regulation performance of urban parks. Second, due to the limitations of orthogonal experimental tables, the investigation into the park landscape design elements and their levels that affect the microclimate of urban blocks is limited.

Therefore, further research can involve simulating the continuous variable changes of the influencing factors to study their relationship with the microclimate of urban blocks. Third, the ENVI-met model in this study used single tree species in the green space of the urban park, which was an ideal situation rather than a real scenario. Further studies are expected to investigate the influence of parks with various plant configurations on the microclimate of urban neighborhoods. Lastly, this study only focused on urban blocks with low-rise and high-rise buildings. It is important to consider more indicators related to urban block morphology, such as floor area ratio, building density, and building layout, in order to clarify the effect of parks on urban blocks with different spatial patterns.

## 5. Conclusions

This study focused on the hot and humid weather in summer in Zhengzhou, a cold region in China. It investigated the impact of landscape design parameters on the temperature difference between the urban park and its surroundings. By combining field measurements, simulations, and quantitative analysis, the study compared three important time points during the day (7:00, 14:00, 19:00) to explore the influence of landscape design parameters on the thermal environment of surrounding neighborhoods with two building heights (18 m, 54 m). Using orthogonal experiments, the interactive effects of landscape design parameters on the temperature difference were analyzed, and a quantitative relationship was established. Finally, a weighted ranking determined the optimal combination, aiming to propose park optimization strategies for enhancing resident comfort and guiding urban forest development.

Based on the analysis, this study summarized the impact mechanisms of parks on the air temperature regulation in surrounding neighborhoods, providing guidance for urban forest development in Zhengzhou. Key research findings are as follows: (1) Landscape design parameters consistently affect the temperature difference between the park and surrounding areas. Factors' variations at midday have the greatest influence on cooling. High-rise building neighborhoods show a larger impact compared to low-rise ones, resulting in temperature reductions of 0.27 °C–0.64 °C for each factor during midday. (2) The influence of landscape design parameters on the temperature difference varies. For low-rise building neighborhoods (18 m), PGA contributes significantly in the morning (50.43%), and PWA is prominent during midday (73.02%) and evening (52.30%). For high-rise building neighborhoods (54 m), PA contributes most in the morning (62.11%), and PWA is prominent during midday (43.99%), with both factors playing crucial roles in the evening, contributing over 80% combined. (3) Weighted ranking of experimental schemes provides optimization strategies for cooling effects in Zhengzhou's urban parks. Recommendations differ for different scenarios. For a 54 m building height, PA of 10 ha–15 ha, PGA at 60%–70%, and LSI above 1.2 are advisable. For an 18 m building height, PA of 5 ha, PGA at 70%–80%, and LSI below 1.2 are recommended. Maintaining a 20% proportion of water bodies significantly enhances the summer thermal environment of surrounding neighborhoods.

**Author Contributions:** Conceptualization, S.X., L.Y. and K.W.; methodology, L.Y. and J.W.; software, J.W.; validation, L.Y. and Y.P.; formal analysis, K.W. and J.W.; investigation, S.X. and L.Y.; resources, K.W.; data curation, S.X. and J.W.; writing—original draft preparation, S.X. and L.Y.; writing—review and editing, S.X., L.Y. and K.W.; visualization, K.W., L.Y. and Y.P.; supervision, S.X.; project administration, S.X.; funding acquisition, S.X. and K.W. All authors have read and agreed to the published version of the manuscript.

**Funding:** This research was funded by the National Natural Science Foundation of China (No. 51808503), the National Natural Science Foundation of China (No. 52208056), the General Research Project of Humanities and Social Science of the Ministry of Education (No. 20YJCZH154), the Key R&D and Promotion Projects in Henan (No. 222102320187), and the Key R&D and Promotion Projects in Henan (No. 232102320178).

**Data Availability Statement:** No new data were created, and data are unavailable due to privacy or ethical restrictions.

**Acknowledgments:** We thank the anonymous reviewers for their valuable comments on an earlier draft of this paper.

**Conflicts of Interest:** The authors declare no conflict of interest.

## Nomenclature

| | |
|---|---|
| LSI | Landscape Shape Index |
| PGA | Percentage of Green Area |
| PA | Park Area |
| PWA | Percentage of Water Area |
| Ta | Air temperature |

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
