# Peer review of "Comparing the Impact of Urban Park Landscape Design Parameters on the Thermal Environment of Surrounding Low-Rise and High-Rise Neighborhoods"

_forests, doi:10.3390/f14081682_

Round 1

Reviewer 1 Report

This study exams the effect of various parks’ landscape design parameters on the thermal environment of surrounding low-rise and high-rise building neighborhoods, thereby proposing the optimal combinations of landscape design parameters of urban parks. This topic is interesting and has practical reference value for urban park planning and design. In my opinion the article is well structured and very well written in all sections. The figures are clear, relevant and informative. The article is also very detailed, thus there are only some minor comments.

Specific comments:

(1) Title: this study mainly focuses on the urban parks’ impact on the thermal environment of different types of neighborhoods, hence the title is a bit too long and not accurate enough. As an example, a possible alternative title could be “Comparing impact of combinations of urban park landscape design parameters on the thermal environment of surrounding low-rise and high-rise neighborhoods”.

(2) In the prototype model of parametric simulation experiment, the greenery is arranged around the water body in the middle. Why was this layout of prototype adopted? Please supplement a necessary and concise explanation.

(3) Could the authors illustrate the field measurement time and explain why such time has been chosen?

(4) In 2.3.2 Software verification, the change trends and characteristics of Ta and RH, which could be shown in Fig. 3, were described. The paragraph dose not agree with what the picture shows.

(5) This study considered the interaction between PA, LSI and PGA in the experimental scheme. Could the authors explain why the interaction between PWA and other landscape design elements was not considered?

(6) Please supplement the limitations of this study.

Reviewer 2 Report

Through orthogonal experiment design and microclimatic numerical simulation, this study investigated the influence of landscape design parameters (Landscape Shape Index, Percentage of Green Area, Park Area and Percentage of Water Area) on the difference of thermal environment within and beyond an urban park during different times. And then the combination of landscape design parameters of urban park, meeting the requirements of microclimate regulation in two types of neighborhoods, was proposed to provide guidance for the planning and construction of urban forest. I recommend a few minor changes:

L120 “The highest average temperature (32.7°C) occurs 119 in June, and the maximum average humidity (75%) is observed in August.” Please confirm this information again.

L185 Why choose these 5 sample points? Is there any difference in the environment around the sample point?

L185 Please supplement the specific information of the field measurement, such as measuring time and duration.

L193 Please analyze the similarity and difference between the measured data and simulated data based on Fig. 3. Are there indicators for quantitative evaluation of validation results?

Fig. 6 Please confirm this picture. Is the largest Ta reduction in case 18? And the smallest in case 13?

In 3.1.2, for the comparison of cooling capacity and range in high-rise and low-rise block, a more specific comparative analysis should be made for the different orientation.

There are still some grammatic errors in the current manuscript. Therefore, moderate editing of English language is required.

Reviewer 3 Report

1.          Why the morphological indicator of LSI in Section 2.2.1 was set to -1.13, 1.2, 1.4 is not clearly known from Table 1, please add explanation.

2.          In section 2.2, please add the concept of the Orthogonal Experiment, the experimental process, and referenced literature.

3.          Table 3 is difficult to understand, such as how to cognitive it with the progress of the orthogonal experiment, and what the simulation cases are after matching with Table 3. Please integrate comment 2 to add expiations and systematically supplement the text and illustrations.

4.          In section 2.3, what are the specific cases for all simulations? How many cases are there? What it looks like in a 3D model? Please integrate comments 2,3 and add expiations.

5.          The data and charts of the evidence that the results explained in lines 284-285 should be supplemented.

6.          In the discussion of section 4, the meaning of each subchapter is more difficult to understand. The existing discussion structure and content seem unsuitable as a section of discussion but should be research results. Therefore, it is recommended that the author revise the article structure, and supply the analysis methods of this part in 2.2 of research methods.

7.          The discussion in section 4 should be a broader discussion of the differences between the results of this study and those of others, the possible limitations of this study, future research suggestions, extended applicable parts, etc.

8.          The descriptions of lines 354-364 do not match those shown in Figure 10. For example, the description in the text indicates that 14:00 has a cooling effect, but the figure showed that it increases the temperature (red). Please check whether it’s an error.

No comment.

Round 2

Reviewer 3 Report

No comment.